# A Multicontinuum-Theory-Based Approach to the Analysis of Fiber-Reinforced Polymer Composites with Degraded Stiffness and Strength Properties Due to Moisture Absorption

Evan Anderson *, Budi Gunawan, James Nicholas, Mathew Ingraham and Bernadette A. Hernandez-Sanchez

Sandia National Laboratories, 1515 Eubank SE, Albuquerque, NM 87125, USA
* Correspondence: evaande@sandia.gov

**Abstract:** Marine energy generation technologies such as wave and tidal power have great potential in meeting the need for renewable energy in the years ahead. Yet, many challenges remain associated with marine-based systems because of the corrosive environment. Conventional materials like metals are subject to rapid corrosive breakdown, crippling the lifespan of structures in such environments. Fiber-reinforced polymer composites offer an appealing alternative in their strength and corrosion resistance, but can experience degradation of mechanical properties as a result of moisture absorption. An investigation is conducted to test the application of a technique for micromechanical analysis of composites, known as multicontinuum theory and demonstrated in past works, as a mechanism for predicting the effects of prolonged moisture absorption on the performance of fiber-reinforced composites. Experimental tensile tests are performed on composite coupons with and without prolonged exposure to a salt water solution to obtain stiffness and strength properties. Multicontinuum theory is applied in conjunction with micromechanical modeling to deduce the effects of moisture absorption on the behavior of constituent materials within the composites. The results are consistent with experimental observations when guided by known mechanisms and trends from previous studies, indicating multicontinuum theory as a potentially effective tool in predicting the long-term performance of composites in marine environments.

**Keywords:** wave energy; composite materials; modeling; failure prediction; property degradation

## 1. Introduction

The renewable energy industry is fluid and rapidly changing in the ever-increasing urgency to build a sustainable network of energy supply and infrastructure. While many unknowns remain about the path ahead, there is little doubt that marine-based renewable energy generation applications, such as wave/tidal power and offshore wind farms, will play an increasingly critical role in the migration to renewable technologies. The vast expanse of the ocean offers tremendous potential for energy production in the form of wave power [1,2]. Tidal, river, and ocean currents provide hydrokinetic energy power and can potentially provide up to 9% of U.S. electricity generation [3–6]. Marine wind resource is also both abundant and high quality in terms of wind speed and continuity [7,8]. Furthermore, world populations are typically concentrated near the coastlines, making marine-based energy sources an attractive option for accessibility and minimization of transmission requirements—an ongoing challenge related to renewables [9–11].

Despite these encouraging aspects, numerous challenges exist to encumber the successful deployment of offshore energy generation technologies. The levelized cost of energy (LCOE) for wave and tidal energy technologies is still relatively high compared with that for utility scale solar and land-based wind energy technologies [12]. One potential means to reduce the LCOE for wave and tidal energy technologies is by utilizing advanced materials that improve the durability and expected life of the marine energy technologies. Conventional materials like steel and other metals are susceptible to rust and break down in a

wet and highly corrosive ocean environment, severely reducing the integrity and lifetime of offshore structures. Fiber-reinforced polymer composite materials (FRPs), with their resistance to corrosion as well as high specific strength and tailorable properties, have been looked to as a promising alternative [13–26]. A caveat, however, is that FRPs have been observed to experience degradation of mechanical properties over time in aqueous environments as a result of water absorption [16–22,27–29]. In order to employ FRPs in the construction of offshore structures with confidence, it is necessary to understand the long-term environmental effects on key mechanical properties in order to accurately predict performance and longevity.

Much work has been carried out in an effort to understand the effects of moisture and seawater absorption on the mechanical properties of FRPs. Several common observations and conclusions that have been found throughout the literature can be highlighted. One such observation is that the tensile strength and shear strength of composite laminates exposed to prolonged moisture, be it pure water, salt water, or sea water, are seen to consistently degrade over time in FRPs [16–19,22]. Stiffness, or Young's modulus of such specimens, can sometimes degrade marginally, but usually to a lesser extent than ultimate strengths [16,22]. It is widely concluded that the degradation in mechanical properties is mainly attributed to breakdown in the polymer resin matrix, and especially in the fiber/matrix interface [16–19,22]. A key mechanism in this is the residual stresses caused by swelling of the polymer matrix as a result of water absorption. The much stiffer fibers remain relatively unaffected by such swelling, causing compressive stress in the matrix regions in the fiber direction [17,19,22]. Prolonged exposure to moisture can sometimes lead to damage and degradation in glass fibers, but to date, there is evidently no such effect on carbon fibers [16–18].

Special findings have been reported in various works that are worth noting. Chen et al. [20] investigated the performance degradation of hybrid composites, which contain multiple kinds of fibers, finding that the tensile strength, wear resistance, and hardness of moisture saturated samples could be improved by using such hybrid systems with the proper balance of volume fractions. Garcia-Espinal et al. [18] compared the performance of saturated composites with several different polymer resins, showing that, although the mechanical properties degraded with moisture saturation in all tested samples, those with epoxy resin reached a point where the degradation subsided, and the properties stabilized. This makes epoxy an attractive choice for polymer resin in marine environments.

The degradation of mechanical properties as a result of moisture/seawater absorption in FRPs exacerbates the already challenging task of predicting failure in composite materials. Because of their microstructure and heterogeneous makeup, composites have numerous distinct modes of failure, with the strength and implication of each depending on the nature and direction of loading. This makes the failure envelope of composites notoriously difficult to define succinctly and complicates failure predictions. Many ultimate failure theories for FRPs have come to be widely accepted, including but not limited to those of Tsai-Wu [30], Puck [31], and LaRC [32,33]. Though these have been well-validated and shown to be appropriate in the proper context, they rely on material-dependent constants that must be empirically determined, and there is no certain and straightforward way of knowing how those constants might be affected by the kind of material degradation phenomena just discussed. Establishing experimental data for materials under long-term moisture exposure is not only inherently time consuming, but must be repeated for each specific composite. This creates a practical barrier for using these established failure theories for structures in marine environments.

Another possible approach is to predict the performance of a composite material system using some fundamental understanding of the microstructural mechanisms driving the behavioral changes the system is undergoing. Such understanding enables modeling of the property degradation in a physically representative way, providing a robust means of predicting the material's behavior. One technique, developed by Garnich et al. [34,35] and referred to as multicontinuum theory (MCT), aims to assess the response of a het-

erogeneous material under loading by decomposing the overall stress and strain of the mixture into stresses and strains in the individual material constituents. This is possible through knowledge of the elastic properties of the mixture as well as those of the individual constituents combined with volume averaging laws. It is then possible to determine a failure index for each constituent, revealing the presence and nature of any potential failure in the material. MCT has been shown to be a powerful and effective predictive tool for ultimate failure in composites [36–38].

The present work is an investigation of the degradation of stiffness and strength properties of FRPs as a result of moisture absorption, using an MCT-based approach. A set of experimental tests were performed on glass/carbon/epoxy hybrid composite coupons to determine elastic and strength properties for dry samples and salt water saturated samples. A modeling scheme was developed using classical laminate theory along with finite element modeling of the microstructural response of FRPs. By combining the experimental data and modeling tools, it is sought to verify whether an MCT-based analysis effectively predicts the response of composites subject to moisture absorption based on what is known from observations in the experiments and from previous work.

## 2. Materials and Methods

### 2.1. Background and Derivation of Multicontinuum Theory

A composite material, such as a fiber-reinforced polymer laminate, is inherently heterogeneous, being composed of multiple constituent materials in a given lamina, and potentially different compositions for each lamina. However, at the structural or design scale, with characteristic lengths much larger than the diameters of fibers or thickness of plies, the material or section properties can often be treated as homogeneous for the purpose of analyzing the overall response of a structure under loading. Doing so is typically attractive for the sake of efficiency. In the context of predicting failure, however, it is important to recognize that, if the constituent materials making up the composite have drastically different mechanical properties, they will have drastically different mechanical responses at the microscale under loading. It is thus advantageous to break down the overall response of the body at a point into the responses for the constituents individually and perform failure analysis based on those constituent responses.

In the present context, a point in a body is not an infinitesimal point as in continuum mechanics or vector calculus, but a finite sub-volume of the body large enough to contain all constituent materials in the vicinity in representative proportions, but small enough to experience a macroscopically uniform state of stress and strain under loading. Often, this concept is referred to as a representative volume element (RVE). Let us limit the scope of the current discussion to the case of mixtures with two distinct constituents, denoted by $\alpha$ and $\beta$. Figure 1 illustrates the concept of an RVE within a fiber-reinforced polymer composite laminate, with $\alpha$ and $\beta$ representing the fiber and resin matrix, respectively.

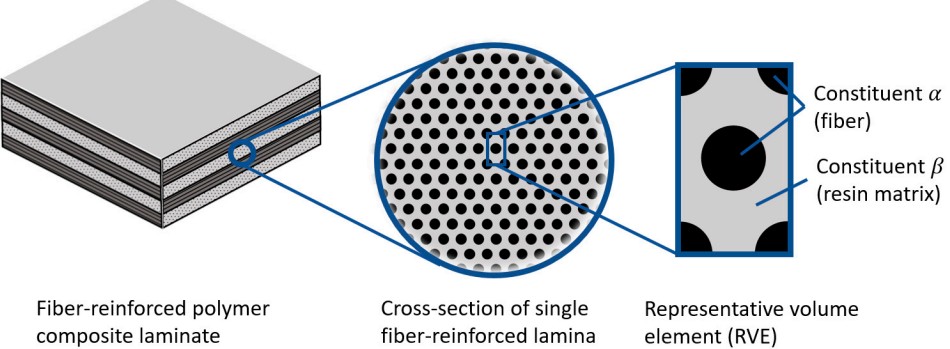

Fiber-reinforced polymer composite laminate

Cross-section of single fiber-reinforced lamina

Representative volume element (RVE)

Constituent $\alpha$ (fiber)

Constituent $\beta$ (resin matrix)

**Figure 1.** Representative volume element within a fiber-reinforced polymer composite laminate.

The fundamental observation is that, under a given loading state, the overall stress and strain at a point, or composite stress $\sigma_c$ and composite strain $\epsilon_c$, are the same quantities volume-averaged over the sub-domain that point represents, denoted $\Omega$. Mathematically,

$$\sigma_c = \frac{\int_\Omega \sigma dV}{V_{tot}} \tag{1}$$

$$\epsilon_c = \frac{\int_\Omega \epsilon dV}{V_{tot}} \tag{2}$$

In Equations (1) and (2), the volume integrals can be split into the domains representing constituents $\alpha$ and $\beta$, with $\Omega_\alpha$ and $\Omega_\beta$, giving

$$\sigma_c = \left(\frac{V_\alpha}{V_{tot}}\right)\frac{\int_{\Omega_\alpha} \sigma dV}{V_\alpha} + \left(\frac{V_\beta}{V_{tot}}\right)\frac{\int_{\Omega_\beta} \sigma dV}{V_\beta} \tag{3}$$

$$\epsilon_c = \left(\frac{V_\alpha}{V_{tot}}\right)\frac{\int_{\Omega_\alpha} \epsilon dV}{V_\alpha} + \left(\frac{V_\beta}{V_{tot}}\right)\frac{\int_{\Omega_\beta} \epsilon dV}{V_\beta} \tag{4}$$

Adopting the following notation for the volume fractions of constituents $\alpha$ and $\beta$, $\phi_\alpha = V_\alpha/V_{tot}$ and $\phi_\beta = V_\beta/V_{tot}$, and Equations (3) and (4) can be rewritten in terms of the volume-averaged stress and strain in constituents $\alpha$ and $\beta$:

$$\sigma_c = \phi_\alpha \sigma_\alpha + \phi_\beta \sigma_\beta \tag{5}$$

$$\epsilon_c = \phi_\alpha \epsilon_\alpha + \phi_\beta \epsilon_\beta \tag{6}$$

This result can be coupled with the knowledge that both constituents, as well as the composite as a whole, have constitutive laws governing the stress–strain relationship. Assuming linear elasticity, these can be expressed with the elastic compliance matrices for the composite, constituent $\alpha$, and constituent $\beta$—$[S_c]$, $[S_\alpha]$, and $[S_\beta]$, respectively:

$$\epsilon_c = [S_c]\sigma_c \tag{7}$$

$$\epsilon_\alpha = [S_\alpha]\sigma_\alpha \tag{8}$$

$$\epsilon_\beta = [S_\beta]\sigma_\beta \tag{9}$$

The composite constitutive law in Equation (7) can be used to obtain the overall response of the body under loading, through analytical or numerical/finite element analysis. Once $\sigma_c$ and $\epsilon_c$ are known, the constituent stresses and strains can be found by combining Equations (5), (6), (8), and (9), sequentially evaluating

$$\sigma_\alpha = \frac{1}{\phi_\alpha}\left([S_\alpha] - [S_\beta]\right)^{-1}\left(\epsilon_c - [S_\beta]\sigma_c\right) \tag{10}$$

$$\epsilon_\alpha = [S_\alpha]\sigma_\alpha \tag{11}$$

$$\epsilon_\beta = \frac{1}{\phi_\beta}(\epsilon_c - \phi_\alpha \epsilon_\alpha) \tag{12}$$

$$\sigma_\beta = [S_\beta]^{-1}\epsilon_\beta \tag{13}$$

The resulting constituent stresses and strains can be used to make assessments regarding failure and maximum loading. How the composite elastic behavior relates to that of the constituents depends strongly on microstructure, and finite element models of RVEs for the material are a powerful tool in establishing that relationship and completing the constitutive laws in Equations (7)–(9). This will be explained and demonstrated further in the next section.

For the present study, the composite structures of interest are carbon–glass hybrid fiber-reinforced test coupons under uniaxial loading. This being the case, it was appropriate to perform coupon-level analysis using classical laminate theory (CLT), followed by MCT analysis on the individual lamina level, to analyze constituent stresses. Figure 2 graphically depicts the workflow of the process.

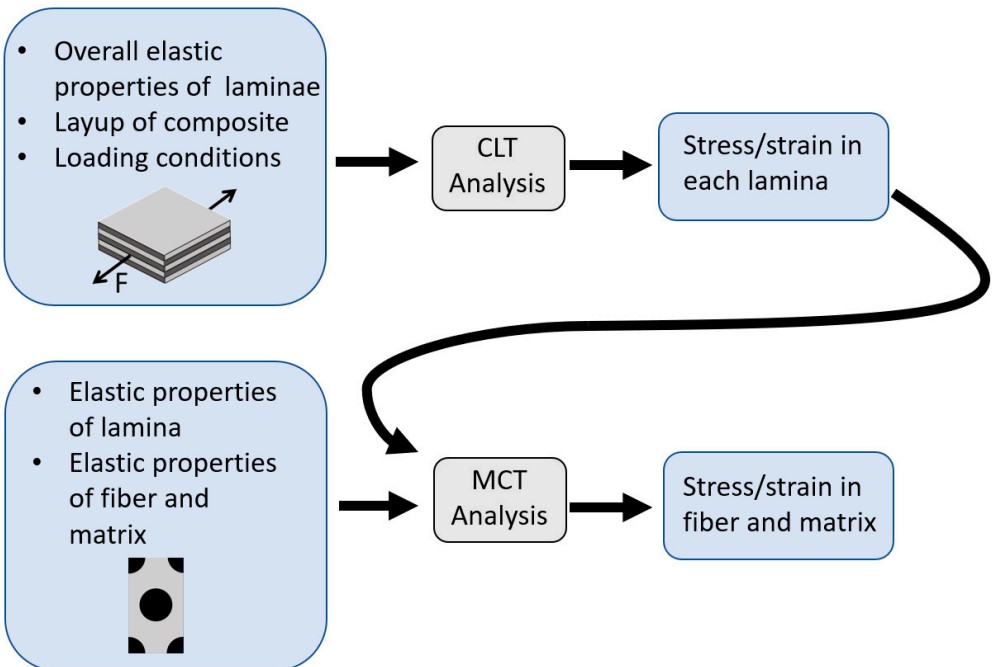

**Figure 2.** Flowchart depicting the two-level process of failure analysis in composite laminate employing classical laminate theory in conjunction with multicontinuum theory.

### 2.2. Experimental Testing

A common technique for joining composite structures is through hole bolting; therefore, a bearing response test based on ASTM D5961-17 was utilized for evaluation of the effects of water absorption on glass/carbon/epoxy hybrid composite coupon samples. The test specimens were machined from a single composite plate of proprietary constituent materials supplied by a marine energy developer. Composite coupons were fabricated in a symmetric 16-layer configuration, [+45° glass, −45° glass, 0° carbon, +45° glass, −45° glass, 0° carbon, +45° glass, −45° glass]$_s$. Two sets of samples were investigated to determine the elastic and strength properties for the dry and salt water saturated conditions. Each set of samples contained five coupon samples.

The salt water solution was held at 77 ± 2 degrees Fahrenheit (25 ± 1.1 degrees Celsius), with a salinity of 34 ppt, using instant ocean sea salt for aquariums. This gives the composition of major ions as shown in Table 1.

Five sets of samples were tested and compared to determine elastic and strength properties for the dry samples and salt water saturated samples. The nominal dimensions used for the test specimens were 135 mm (length) × 36 +/− 1 mm (width) × 2 to 4 mm (height). The samples, a typical example of which is shown in Figure 3, were fabricated with a through bolt configuration with two plates on opposing sides. The plates and connecters are 316 stainless steel. The tests were conducted on a MTS Model 312.21 tensile test stand using a model 661.21A-03 load cell and hydraulic grippers. Each test lasted for a few minutes, until the sample failed.

The bearing test configuration was chosen rather than standard tensile tests based on the needs of the marine industry. A survey of industry leaders was completed prior to testing to determine the most common joining methods used. The through bolt lapped joint was among the most used and most concerning joints owing to the strong stress concentra-

tion at the bolt hole in the laminate. The ASTM standard test method for bearing response of polymer matrix composite laminates is the most relevant to this type of connection and provides the industry with information that can directly correlate to application.

**Table 1.** Major ion composition of the salt water solution used to submerge test coupons.

| Major Ion | | Salt Composition at 34 ppt Salinity (mg/L) |
|---|---|---|
| Chloride | $Cl^-$ | 18,740 |
| Sodium | $Na^+$ | 10,454 |
| Sulfate | $SO_4^{2-}$ | 2631 |
| Magnesium | $Mg^{2+}$ | 1256 |
| Calcium | $Ca^{2+}$ | 400 |
| Potassium | $K^+$ | 401 |
| Bicarbonate | $HCO_3^-$ | 194 |
| Boron | $B^{3+}$ | 6 |
| Strontium | $Sr^2$ | 7.5 |

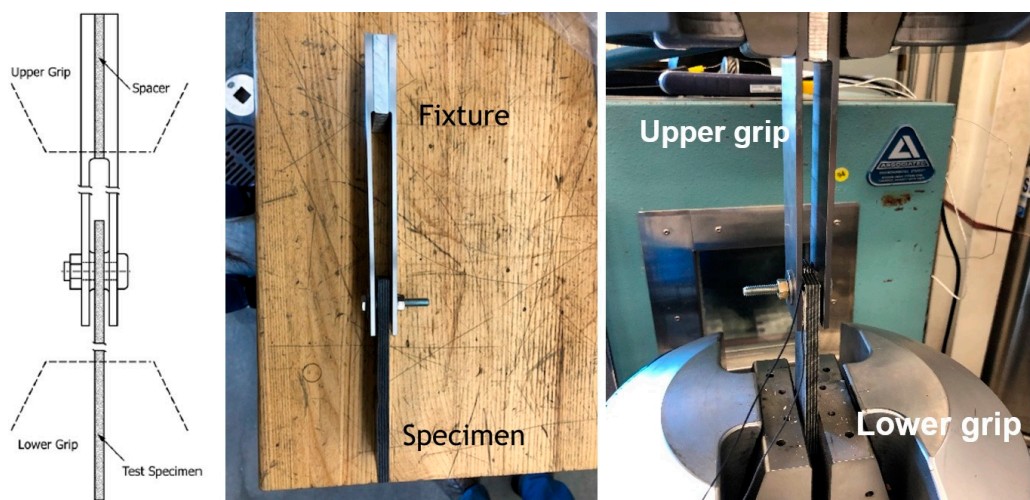

**Figure 3.** Example of the typical bearing test configuration used for analyzing mechanical properties.

Experimental tests such as these play a critical role in the modeling procedure and analysis that follows. MCT-based failure analysis relies on an understanding of the stiffness and strength properties of individual constituents. However, these can be highly dependent on many variables including environmental conditions and manufacturing quality. Comparisons of test results between dry (control) and conditioned samples allow for the calibration of the parameters in these models for different manufacturers and configuration types. More details of this process are given in the following section.

### 2.3. Modeling and Analysis

### 2.3.1. Summary of Modeling Procedure

The results of the experimental testing were used along with published data in the literature to perform the modeling and analysis phase of this work. The objective was to gain a meaningful understanding of the micromechanical mechanisms of moisture-induced stiffness and strength degradation and verify the effectiveness of MCT analysis in accounting for these effects. The main steps of the modeling investigation can be summarized as follows:

1.  Deduce the elastic properties of the individual glass and carbon fiber lamina within the dry and saturated test coupons, given the experimental stiffness measurements, the layup and volume fractions, and properties known from the literature. This was performed using classical laminate theory within gradient-based optimization to match measured laminate properties. A finite element model of an RVE within a lamina of fiber-reinforced polymer composites was used to obtain initial estimates of properties.

2.  Deduce the actual effective elastic properties of glass and carbon fibers and epoxy resin as constituents in the dry coupons based on published data along with the determined lamina properties from step 1. This was performed using the finite element RVE model within gradient-based optimization. Find the effective degraded stiffness of epoxy resin in saturated coupons under two separate assumptions: (a) the stiffness of epoxy degrades throughout the RVE domain and (b) the stiffness of epoxy degrades only in the thin layer forming the fiber/matrix interface.

3.  Deduce the effective in situ strength of the epoxy matrix based on the ultimate strength results of the test coupons using MCT analysis, for both dry and saturated coupons under the two assumptions mentioned in step 2. This was performed using classical laminate theory, employing all lamina and constituent elastic properties found in previous steps into the MCT analysis.

4.  Examine the effect of moisture-induced swelling/expansion in the epoxy resin matrix on the MCT-based results for constituent stresses. This was performed using modified forms of classical laminate theory and MCT augmented to account for expansion phenomena, along with simulation of composite expansion behavior with the finite element RVE model.

Further details and explanations of each of these steps are given in the following sections, followed by results and discussion of implications.

### 2.3.2. Elastic Properties of Glass/Carbon Lamina

As described in Section 2.1, the elastic behaviors of a composite region overall and for each individual constituent material are necessary ingredients for MCT analysis. In this case, the test coupons were carbon/glass fiber epoxy hybrid laminates for which tensile tests were performed to obtain stiffness and strength properties, as explained in Section 2.2. To deduce meaningful information through modeling from these tests, the elastic properties of the individual carbon- and glass-reinforced lamina must be verified, followed by those of the carbon and glass fibers and epoxy resin matrix on the microscale. There are more than sufficient data published in the literature to obtain a solid estimate for these properties. The strategy was to begin with such an estimate and fine-tune the elastic properties so that the predicted laminate stiffness matched the average experimentally measured results through gradient-based optimization.

Approximate values for the elastic properties of carbon fiber, glass fiber, and epoxy resin matrix based on published data are shown in Table 2. A finite element model of an RVE within a lamina of fiber-reinforced polymer composite was constructed using Abaqus commercial finite element modeling software in order to predict the overall elastic properties of the carbon-reinforced lamina and the glass-reinforced lamina based on the properties in Table 2. The RVE model was built of eight-node linear hexahedral elements with incompatible modes, assuming a hexagonal packed fiber arrangement and a fiber volume fraction of 0.4 to match that of the test coupons. A thin layer of elements was placed around the fibers in the model to represent the fiber/matrix interface, which could be given independent material properties. Overall lamina properties were obtained by applying macroscopically uniform states of stress to the RVE model while enforcing fully generalized periodic boundary conditions in all directions, and taking volume-averages of the resulting strain. Figure 4 shows a graphic depiction of the RVE model and periodic response under loading.

**Table 2.** Approximate values for elastic properties of constituent materials taken from a survey of the literature [39–41].

| Constituent Material | Approximate Published Elastic Properties |
|---|---|
| Carbon Fiber (transversely isotropic) | $E_1 \approx 270$ GPa<br>$E_2 \approx 20$ GPa<br>$G_{12} \approx 70$ GPa<br>$\nu_{12} \approx 0.25$<br>$\nu_{23} \approx 0.7$ |
| Glass Fiber (fully isotropic) | $E \approx 70$ GPa<br>$\nu \approx 0.2$ |
| Epoxy Resin (fully isotropic) | $E \approx 3.5$ GPa<br>$\nu \approx 0.35$ |

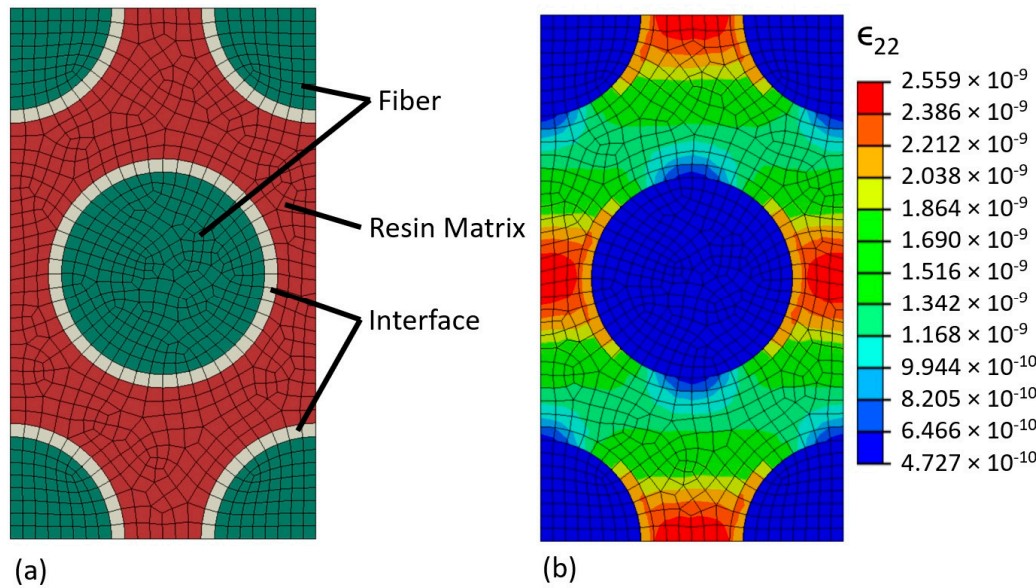

(a)            (b)

**Figure 4.** (**a**) Cross-sectional depiction of finite element RVE model of fiber-reinforced polymer lamina and (**b**) periodic strain response under transverse loading.

The RVE model was constructed and checked for correctness and functionality in Abaqus, though the bulk of the analysis was performed with ASenD3D, an open-source code for finite element modeling sensitivity analysis of structures. Formerly known as AStrO and demonstrated in previous works [42–44], ASenD3D performs high-fidelity analysis of the thermal and elastic response of 3D structures and computes sensitivities of structural objectives using the adjoint method. A publicly available repository for the tool is under construction at the time of this publication and is accessible at https://github.com/MSDOToolz/ASenD3D (accessed on 31 January 2023). Using the fiber and epoxy matrix properties from Table 2, the initial estimates for the elastic properties of the carbon and glass lamina are presented in Table 3.

The lamina properties from Table 3 were then modified to fit the average overall laminate stiffnesses measured in the dry test coupons. Following classical laminate theory, the overall mechanical response of a layered composite laminate subject to in-plane loading can be defined by relating the applied forces per unit length, $N = [N_1, N_2, N_{12}]^T$, and moments per unit length, $M = [M_1, M_2, M_{12}]^T$, to the strains and curvatures in a reference plane, $\epsilon_0 = [\epsilon_1, \epsilon_2, \gamma_{12}]^T$ and $\kappa = [\kappa_1, \kappa_2, \kappa_{12}]^T$, respectively, as follows:

$$\begin{Bmatrix} N \\ M \end{Bmatrix} = \begin{bmatrix} [A] & [B] \\ [B] & [D] \end{bmatrix} \begin{Bmatrix} \epsilon_0 \\ \kappa \end{Bmatrix} \tag{14}$$

The matrices $[A]$, $[B]$, and $[D]$ are formed from the thickness, in-plane orientation, and elastic properties of each lamina through the laminate thickness.

**Table 3.** Elastic properties of carbon and glass lamina estimated from finite element RVE model and constituent properties from the literature in Table 2.

| Fiber Type | Initial Estimated Lamina Elastic Properties |
|---|---|
| Carbon | $E_1$ = 109.9 GPa<br>$E_2$ = 6.48 GPa<br>$G_{12}$ = 2.91 GPa<br>$\nu_{12}$ = 0.30<br>$\nu_{23}$ = 0.53 |
| Glass | $E_1$ = 30.0 GPa<br>$E_2$ = 7.54 GPa<br>$G_{12}$ = 2.79 GPa<br>$\nu_{12}$ = 0.28<br>$\nu_{23}$ = 0.45 |

A tool was built in MATLAB to evaluate the overall longitudinal and transverse stiffness of a composite laminate given a list of properties for each lamina using this approach. Then, an optimization script was set up to find the best least-squares fit to match the predicted laminate stiffnesses with the average measured results by changing the lamina elastic properties given in Table 3. The optimization algorithm used was fmincon, a gradient-based optimizer built into matlab with the fitness, or objective function, defined as

$$f_{obj} = \mu_1 \left( E_{1,predicted} - E_{1,measured} \right)^2 + \mu_2 \left( E_{2,predicted} - E_{2,measured} \right)^2 \tag{15}$$

where $\mu_1$ and $\mu_2$ are constants set at 1 and 25, respectively, based on the approximate magnitude of the longitudinal stiffness relative to that of transverse stiffness. The results were taken as the effective lamina stiffnesses for the dry coupons.

For the saturated coupons, the degradation in longitudinal stiffness was minimal compared with the degradation in transverse stiffness. With that observation along with patterns noted in the literature, it was assumed that the degradation in properties was mainly attributed to breakdown in either the matrix or fiber/matrix interface, and that only the properties $E_2$ and $G_{12}$ were effectively reduced in the carbon and glass lamina. That factor of reduction was found again with optimization, this time holding $E_1$, $\nu_{12}$, and $\nu_{23}$ in the lamina constant, and fitting to transverse stiffness only, with the objective

$$f_{obj} = \left( E_{2,predicted} - E_{2,measured} \right)^2 \tag{16}$$

With effective elastic properties established for the carbon and glass lamina in both dry and saturated coupons, it was possible to proceed to step 2, detailed in the next section.

2.3.3. Constituent Elastic Properties for Glass/Carbon Fiber and Epoxy Resin

Just as the elastic properties of individual lamina in a composite can be inferred from those of the laminate as a whole, the elastic properties of the fibers and resin matrix can be inferred from the lamina properties. In the previous step, the finite element RVE model was used to obtain an initial estimate for lamina properties as a starting point in the optimization. For the present step, the RVE model analysis is the core operation of the optimization, which evaluates the effective, or volume-averaged, elastic properties of the glass and carbon lamina given a set of values for the properties of the fibers and resin matrix. We adopt the notation for volume-averaged stress and strain:

$$\overline{\sigma_i} = \frac{\int_\Omega \sigma_i dV}{V_{tot}} \tag{17}$$

$$\overline{\epsilon_i} = \frac{\int_\Omega \epsilon_i dV}{V_{tot}} \tag{18}$$

Subsequently, for the effective, volume-averaged Young's moduli, shear moduli, and Poisson's ratio for an RVE/lamina, we adopt the following notation:

$$\overline{E_i} = \frac{\overline{\sigma_i}}{\overline{\epsilon_i}} \qquad (under\ normal\ stress\ in\ the\ i\ direction) \tag{19}$$

$$\overline{G_{ij}} = \frac{\overline{\tau_{ij}}}{\overline{\gamma_{ij}}} \qquad (under\ shear\ stress\ in\ the\ ij\ direction \tag{20}$$

$$\overline{\nu_{ij}} = \frac{-\overline{\epsilon_j}}{\overline{\epsilon_i}} \qquad (under\ normal\ stress\ in\ the\ i\ direction) \tag{21}$$

In this phase of optimization, the effective elastic properties for the carbon and glass lamina predicted by the RVE model, namely, volume-averaged $E_1$, $E_2$, $G_{12}$, $\nu_{12}$, and $\nu_{23}$, were tailored to match the final values obtained in the previous phase, by modifying the elastic properties of the carbon fiber, glass fiber, and resin matrix. The objective function was defined as a weighted sum of squares of residuals:

$$f_{obj} = \mu_1\left(\overline{E_1} - E_{1,eff}\right)^2 + \mu_2\left(\overline{E_2} - E_{2,eff}\right)^2 \dots \quad \begin{matrix}(for\ all\ elastic\ properties\ of \\ carbon\ and\ glass\ lamina)\end{matrix} \tag{22}$$

Again, $\mu_1$ and $\mu_2$ were scaled appropriately for the magnitudes of elastic moduli compared with Poisson's ratio. The structural analysis tool AStrO has the functionality to evaluate the exact sensitivities of this objective function with respect to all fiber/matrix elastic properties in the time of a single finite element analysis at each phase, making for efficient gradient-based optimization.

After obtaining the best-fit solution for the fiber and matrix elastic properties in the dry coupons, the degraded Young's modulus in the resin matrix as a result of moisture absorption was found similarly by matching the transverse modulus in the saturated lamina to the effective transverse modulus in the RVE model. This was repeated twice, each time with a different assumption. In the first run, Young's modulus was degraded in the resin matrix throughout the whole RVE domain, until the effective lamina stiffness matched what was found in the previous step. In the second run, the modulus was only reduced in the fiber/matrix interface layer around the fiber, illustrated in Figure 4. For both runs, the properties of both the carbon and glass fibers were held constant, as it is assumed that the fibers themselves do not sustain any significant damage due to water absorption.

### 2.3.4. Failure Strength for Epoxy Resin

With the elastic properties verified for the carbon- and glass-reinforced lamina, as well as for the individual fibers and epoxy matrix, it was then possible to analyze the constituent stress response within the test coupons at the point of maximum failure. Of particular interest for the present study was the effective strength of the epoxy matrix for saturated coupons as compared with dry coupons. The process for deducing the state of stress in the constituents within a composite coupon under a given loading is as follows:

1.  Form the ABD matrix shown in Equation (14) given the composite layup and the properties of each lamina.
2.  Find the response of the composite under the given loading vector, $[N, M]^T$, by solving Equation (14) for the strains and curvatures $[\epsilon_0, \kappa_0]$ at the reference plane.
3.  For every lamina through the thickness:
    a.  Find the in-plane lamina strain from the composite strains and curvatures and the corresponding stress in the lamina by

$$\epsilon = \epsilon_0 + z\kappa \tag{23}$$

$$\sigma = [Q]\epsilon \qquad (24)$$

b.   Find the stress and strain in the constituents of the lamina with the MCT equations, Equations (10)–(13).

Evaluating the stress in the fiber and matrix constituents in the composite under the loading at failure gives a means of deducing the effective strength of those constituents. In the present work, the Von Mises stress was used as a strength metric for the epoxy matrix, in comparisons between dry and saturate composite coupons. The above process was repeated for the dry coupons as well as the saturated coupons under the two different assumptions described in the previous section, in order to compare the implications of assuming stiffness degradation throughout the matrix as opposed to degradation in the fiber/matrix interface only.

2.3.5. Effect of Moisture-Induced Expansion

To examine the influence of expansion in the epoxy matrix due to moisture absorption, the process in the previous section was repeated, but with modified versions of composite laminate theory and MCT to accommodate the expansion effect. A general principle for the mechanics of elastic bodies under the influence of some expansion effect such as moisture or thermal expansion is that the total strain at a point potentially has a component due to applied stress and another component due to expansion, as shown:

$$\epsilon_{total} = \epsilon_\sigma + \epsilon_{exp} \qquad (25)$$

Assuming linear elasticity and linear expansion:

$$\epsilon_{total} = [S]\sigma + c\lambda \qquad (26)$$

where $c$ is the concentration of the expansion-driving species and $\lambda$ is an expansion coefficient vector. By extension, the stress at a point under the influence of expansion can be written as follows:

$$\sigma = [S]^{-1}(\epsilon_{total} - c\lambda) \qquad (27)$$

Or, for the case of plane stress where $\sigma' = [\sigma_1, \sigma_2, \tau_{12}]^T$ and $\epsilon' = [\epsilon_1, \epsilon_2, \gamma_{12}]^T$, we can express stress in terms of the reduced $3 \times 3$ stiffness matrix $[Q]$ and the reduced expansion coefficient vector $\lambda'$, as follows:

$$\sigma' = [Q](\epsilon' - c\lambda') \qquad (28)$$

Returning now to the derivation of forces and moments per unit length in a composite lamina under in-plane loading, the expansion effect works as follows:

$$N = \int_{z-}^{z+} \sigma' dz = \int_{z-}^{z+} [Q](\epsilon' - c\lambda') dz \qquad (29)$$

$$M = \int_{z-}^{z+} z\sigma' dz = \int_{z-}^{z+} z[Q](\epsilon' - c\lambda') dz \qquad (30)$$

Expressing the total strain in terms of strain and curvatures at the midplane,

$$\epsilon' = \epsilon'_0 + z\kappa \qquad (31)$$

Equations (29) and (30) can then be rewritten:

$$N = \int_{z-}^{z+} [Q](\epsilon'_0 + z\kappa - c\lambda') dz \qquad (32)$$

$$M = \int_{z-}^{z+} z[Q](\epsilon'_0 + z\kappa - c\lambda') dz \qquad (33)$$

Broken into a summation over all the lamina through the thickness,

$$N = \left( \sum_{i=1}^{n+1} \int_{z_i}^{z_{i+1}} [Q]_i dz \right) \epsilon_0' + \left( \sum_{i=1}^{n+1} \int_{z_i}^{z_{i+1}} z[Q]_i dz \right) \kappa - \sum_{i=1}^{n+1} \int_{z_i}^{z_{i+1}} c[Q]_i \lambda_i' dz \quad (34)$$

$$M = \left( \sum_{i=1}^{n+1} \int_{z_i}^{z_{i+1}} z[Q]_i dz \right) \epsilon_0' + \left( \sum_{i=1}^{n+1} \int_{z_i}^{z_{i+1}} z^2 [Q]_i dz \right) \kappa - \sum_{i=1}^{n+1} \int_{z_i}^{z_{i+1}} cz[Q]_i \lambda_i' dz \quad (35)$$

The first two summation terms in Equations (34) and (35) are none other than the $[A]$, $[B]$, and $[D]$ matrices that are always present in classical laminate theory. If we now define the expansion load component vectors as

$$N_\lambda = \sum_{i=1}^{n+1} \int_{z_i}^{z_{i+1}} c[Q]_i \lambda_i' dz \quad (36)$$

$$M_\lambda = \sum_{i=1}^{n+1} \int_{z_i}^{z_{i+1}} cz[Q]_i \lambda_i' dz \quad (37)$$

Then, the augmented response of the composite can be written in matrix form as

$$\left\{ \begin{matrix} N \\ M \end{matrix} \right\} = \begin{bmatrix} [A] & [B] \\ [B] & [D] \end{bmatrix} \left\{ \begin{matrix} \epsilon_0 \\ \kappa \end{matrix} \right\} - \left\{ \begin{matrix} N_\lambda \\ M_\lambda \end{matrix} \right\} \quad (38)$$

By first constructing the expansion load vector from the stiffness and expansion properties of each lamina along with the concentration of the driving species, and adding it to the applied load before solving for the reference plane strains and moments, the effect of expansion can be accounted for in the overall composite response.

Similarly, the equations governing MCT can be modified to account for expansion phenomena. Equations (7)–(9) from Section 2.1 become

$$\epsilon_c = [S_c] \sigma_c + c \lambda_c \quad (39)$$

$$\epsilon_\alpha = [S_\alpha] \sigma_\alpha + c \lambda_\alpha \quad (40)$$

$$\epsilon_\beta = [S_\beta] \sigma_\beta + c \lambda_\beta \quad (41)$$

Consequently, Equations (10)–(13) extracting the constituent stress and strain within a lamina become

$$\sigma_\alpha = ([S_\alpha] - [S_\beta])^{-1} \left( \frac{1}{\phi_\alpha} (\epsilon_c - [S_\beta] \sigma_c - \phi_\beta c \lambda_\beta) - c \lambda_\alpha \right) \quad (42)$$

$$\epsilon_\alpha = [S_\alpha] \sigma_\alpha + c \lambda_\alpha \quad (43)$$

$$\epsilon_\beta = \frac{1}{\phi_\beta} (\epsilon_c - \phi_\alpha (\epsilon_\alpha + c \lambda_\alpha)) \quad (44)$$

$$\sigma_\beta = [S_\beta]^{-1} (\epsilon_\beta - c \lambda_\beta) \quad (45)$$

Using these modified forms of CLT and MCT, it was possible to examine the potential effects of moisture-induced expansion on the constituent stresses within the composite coupons under critical loading, as well as the implications that may have. The finite element RVE model was used to obtain the expansion coefficients of the carbon and glass lamina for a given expansion coefficient for the epoxy matrix, assuming no expansion coefficient

in the fibers themselves. The following section contains the results and discussion of the experiments and modeling investigations just described.

## 3. Results

### 3.1. Experimental Results

The numerical results of the tensile tests for modulus, ultimate tensile strength, and maximum strain of the dry and moisture saturated composite coupons are presented in Table 4. Table 5 shows the modulus, strength, and moisture content averaged for all dry and saturated samples. The degradation of properties was considerably more acute in the transverse direction than in the longitudinal direction.

Figure 5 shows images of the composite coupons after testing. Most samples show distinct visible failure patterns in the 45° direction, indicating progressive matrix failure in the glass laminae prior to ultimate failure. Figure 6 shows force–displacement curves for the dry and saturated coupons from initial loading to ultimate failure. The progressive matrix failure is evident in the force–displacement curves as well, by the distinct change in the slope of the curves occurring at around 3500 lb. for all coupon samples.

**Table 4.** Tensile test results for dry and moisture saturated test coupons.

| Coupon | Percent Moisture | Longitudinal Direction | | | Transverse Direction | | |
| | | E (GPa) | UTS (MPa) | Percent Strain | E (GPa) | UTS (MPa) | Percent Strain |
|---|---|---|---|---|---|---|---|
| 1 | 0 | 56.1 | 786 | 1.38 | 10.7 | 98.3 | 3.17 |
| 2 | 0 | 54.8 | 773 | 1.40 | 9.02 | 83.3 | 3.26 |
| 3 | 0 | 54.1 | 792 | 1.43 | 9.96 | 95.3 | 3.67 |
| 4 | 0 | 53.7 | 774 | 1.36 | 8.91 | 83.9 | 3.69 |
| 5 | 0 | 56.5 | 733 | 1.29 | 9.69 | 77.8 | 3.54 |
| 6 | 1.2 | 58.3 | 787 | 1.33 | 8.54 | 68.3 | 1.84 |
| 7 | 1.33 | 55.3 | 725 | 1.30 | 7.79 | 58.9 | 1.84 |
| 8 | 1.1 | 52.1 | 691 | 1.31 | 8.62 | 68.0 | 1.92 |
| 9 | 1.2 | 53.1 | 712 | 1.30 | 8.18 | 60.5 | 1.82 |
| 10 | 0.34 | 57.9 | 695 | 1.15 | 8.05 | 63.6 | 2.05 |

**Table 5.** Key average results for dry and moisture saturated test coupons.

| Condition | Avg. Moisture (%) | Longitudinal Direction | | Transverse Direction | |
| | | Avg. E (GPa) | Avg. UTS (MPa) | Avg. E (GPa) | Avg. UTS (MPa) |
|---|---|---|---|---|---|
| Dry | 0 | 55.0 | 772 | 9.65 | 87.7 |
| Saturated | 1.034 | 55.3 | 722 | 8.24 | 63.9 |

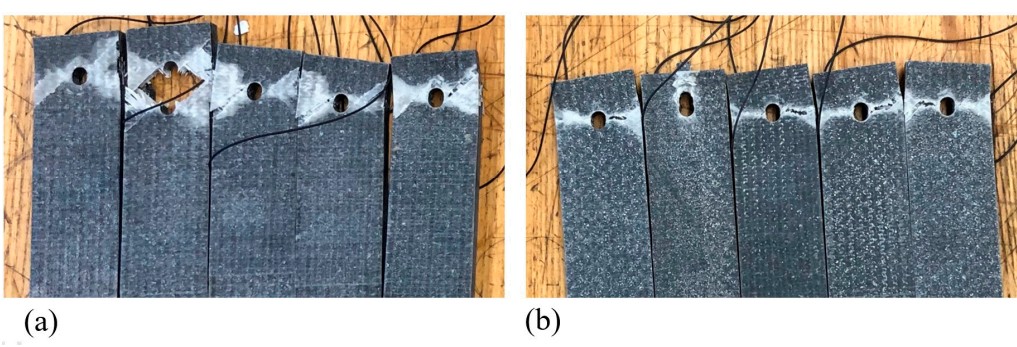

(a)        (b)

**Figure 5.** (**a**) Dry coupons after longitudinal tensile tests. (**b**) Saturated coupons after longitudinal tests.

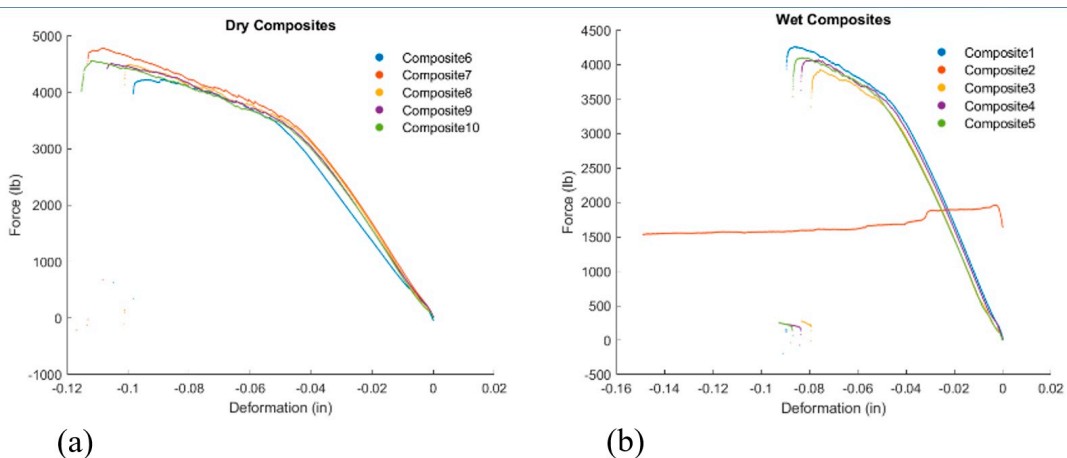

**Figure 6.** Force-displacement curves in longitudinal loading for (**a**) dry coupons and (**b**) saturated coupons.

### 3.2. Modeling Results

The elastic properties for the carbon-reinforced lamina and glass-reinforced lamina obtained from the best-fit optimization in step 1 of the modeling investigation are shown in Table 6.

**Table 6.** Elastic properties for carbon and glass lamina obtained from the optimization process, compared with initially estimated values from the RVE model using published fiber and matrix properties.

| Lamina | Estimated Properties | Optimized Dry Properties | Optimized Saturated Properties |
|---|---|---|---|
| Carbon | $E_1$ = 109.9 GPa<br>$E_2$ = 6.480 GPa<br>$G_{12}$ = 2.910 GPa<br>$\nu_{12}$ = 0.3000 | $E_1$ = 110.1 GPa<br>$E_2$ = 6.292 GPa<br>$G_{12}$ = 2.910 GPa<br>$\nu_{12}$ = 0.3013 | $E_1$ = 110.1 GPa<br>$E_2$ = 4.785 GPa<br>$G_{12}$ = 2.213 GPa<br>$\nu_{12}$ = 0.3013 |
| Glass | $E_1$ = 30.00 GPa<br>$E_2$ = 7.540 GPa<br>$G_{12}$ = 2.790 GPa<br>$\nu_{12}$ = 0.2800 | $E_1$ = 29.75 GPa<br>$E_2$ = 7.235 GPa<br>$G_{12}$ = 2.812 GPa<br>$\nu_{12}$ = 0.2729 | $E_1$ = 29.75 GPa<br>$E_2$ = 5.501 GPa<br>$G_{12}$ = 2.138 GPa<br>$\nu_{12}$ = 0.2729 |

Similarly, the final properties of the constituent materials in dry coupons inferred from the optimization in step 2 of the modeling investigation are presented in Table 7.

**Table 7.** Elastic properties for carbon and glass fibers and epoxy matrix obtained from the optimization process, compared with values estimated from the literature.

| Constituent | Estimated Properties | Optimized Dry Properties |
|---|---|---|
| Carbon Fiber | $E_1$ = 270.0 GPa<br>$E_2$ = 20.00 GPa<br>$G_{12}$ = 70.00 GPa<br>$\nu_{12}$ = 0.2500<br>$\nu_{23}$ = 0.7000 | $E_1$ = 270.9 GPa<br>$E_2$ = 20.16 GPa<br>$G_{12}$ = 70.00 GPa<br>$\nu_{12}$ = 0.2569<br>$\nu_{23}$ = 0.6800 |
| Glass Fiber | $E$ = 70.00 GPa<br>$\nu$ = 0.2000 | $E$ = 69.41 GPa<br>$\nu$ = 0.1959 |
| Epoxy Matrix | $E$ = 3.500 GPa<br>$\nu$ = 0.3500 | $E$ = 3.390 GPa<br>$\nu$ = 0.3392 |

After obtaining the properties for the dry coupons, the degraded Young's modulus in epoxy matrix as a result of water absorption was found for the two assumptions described in Section 2.3:

(a)    Degradation of modulus throughout the matrix: $E$ = 2.480 GPa
(b)    Degradation of modulus at the fiber/matrix interface only: $E$ = 1.125 GPa

Having verified the necessary elastic properties for lamina and constituents, it was possible to perform the MCT analysis on the coupons at critical loading. Table 8 presents the maximum Von Mises stress in each constituent material of the coupons at the point of ultimate failure in the longitudinal and transverse directions, according to the MCT analysis.

**Table 8.** Maximum Von Mises stress (Pa) occurring in each constituent material at the loading of ultimate failure under various loading, conditions, and assumptions.

| | Condition/Assumption | Carbon Fiber | Glass Fiber | Epoxy Matrix |
|---|---|---|---|---|
| **Longitudinal Loading** | Dry | $3.7938 \times 10^9$ | $2.7767 \times 10^8$ | $7.6202 \times 10^7$ |
| | Saturated, degradation throughout matrix | $3.6351 \times 10^9$ | $2.3375 \times 10^8$ | $5.5196 \times 10^7$ |
| | Saturated, degradation at interface only | $3.6154 \times 10^9$ | $2.1763 \times 10^8$ | $7.6362 \times 10^7$ |
| **Transverse Loading** | Dry | $2.3801 \times 10^8$ | $2.8578 \times 10^8$ | $4.4383 \times 10^7$ |
| | Saturated, degradation throughout matrix | $1.9191 \times 10^8$ | $2.4791 \times 10^8$ | $2.7969 \times 10^7$ |
| | Saturated, degradation at interface only | $1.8510 \times 10^8$ | $2.4207 \times 10^8$ | $4.0792 \times 10^7$ |

## 4. Discussion

Figure 5 shows images of the composite coupons after testing. All specimens failed in the composite, not the fastener. For all dry test specimens, the failure mode is similar, as seen in Figure 5a. These failures show both laminate tear out failure and lateral (net tension) failure in the composite, failure code M(TL)1I. The far right and far left specimens show evidence of bearing failure, B1I, but the M(TL)1I failure mode is dominant. All dry samples show distinct visible failure patterns in the 45° direction, indicating progressive matrix failure in the glass laminae prior to ultimate failure. The saturated samples, Figure 5b, show typical ultimate failure in accordance with the common combination of bearing and lateral failure mode, M(LB)1I, except for one. The outlier failure mode is B1I, the laminate bearing failure type. The dominate common failure for the saturated specimen is L1I.

Overall, the estimated values of elastic properties of lamina came close to predicting the experimentally observed coupon stiffnesses when incorporated into classical composite laminate theory, and most of the properties changed only minimally in the optimization process. Because the carbon lamina were all oriented directly in the longitudinal direction, and the loading applied was purely in the longitudinal and transverse directions, the longitudinal shear modulus $G_{12}$ had virtually no influence on the composite response, and it remained at the estimated value. For the saturated coupons, $G_{12}$ was assumed to scale down proportionally with transverse modulus $E_2$, as the two properties are similarly matrix-driven in FRPs. The largest changes occurred in the transvers moduli in both lamina types. The observed reduction in transverse stiffness of the coupons corresponded to about a 25% reduction in the transverse moduli of the lamina. With the elastic properties of the constituent materials in the dry coupons, changes were again minimal overall, with the greatest change being in the Young's modulus for the epoxy matrix.

Given the layup of the coupons, it is expected that the ultimate strength in the longitudinal direction should be dictated by the carbon fiber, and that in the transverse direction by the epoxy matrix, as there are no fibers running directly in the transverse direction in these coupons. Published data for the ultimate strength of carbon fiber run mainly in the vicinity of 3.5 GPa [45], which is in close agreement with the calculated results under longitudinal loading, between 3.6 and 3.8 GPa. The strength of epoxy resin is known to range from about 35 to 50 MPa [46], a range also consistent with the calculated result for transverse

loading. Glass fibers have an ultimate strength around 3.3 GPa [47], which, according to these results, is not even remotely approached in any test.

Under the assumption that the absorption of moisture leads to degradation of epoxy matrix stiffness throughout the volume, the results indicate nearly a 40% reduction in matrix strength. Assuming the stiffness degrades at the fiber/matrix interface only, however, leads to only a marginal change in the calculated strength by about 8%. Either assumption could potentially explain the reduction in transverse strength and stiffness of the coupons alone, but some additional insight can be gained looking at the force–displacement curves produced by the experiment, in Figure 6. In the longitudinal tensile test, a yield point can be clearly seen midway through the curve where the slope decreases substantially, corresponding to the point where the matrix begins cracking in the glass cross plies in the fiber direction, reducing the stiffness of the coupon. Indeed, the results indicate that, at failure in longitudinal loading, the matrix has exceeded its ultimate strength, as deduced from the transverse loading case. It is particularly worth noting that the yield point occurs at roughly the same loading for both the dry and saturated coupons. If the epoxy matrix underwent a significant reduction in strength due to moisture absorption, as the first assumption implies, the yield point should occur at a lower loading for the saturated composites than for the dry composites. Using the second assumption, which is more supported by previous work, leads to predictions more consistent with these experimental observations.

An interesting observation was seen in studying the effect of moisture-induced expansion in the matrix as well. Simulation with the RVE finite element model revealed that, for a given coefficient of moisture expansion in the epoxy matrix $\lambda_m$, the corresponding expansion coefficient vectors for the carbon and glass lamina scaled as follows:

$$\boldsymbol{\lambda}_c = \lambda_m [0.01813, \ 0.7776, \ 0.7776, \ 0, \ 0, \ 0]^T \tag{46}$$

$$\boldsymbol{\lambda}_g = \lambda_m [0.06779, \ 0.7613, \ 0.7613, \ 0, \ 0, \ 0]^T \tag{47}$$

A publication by Lai et al. [47] documented a thorough study of the behavior of epoxy resin with moisture absorption. Their results showed some nonlinearity in the hygroscopic strain as a function of moisture concentration, but in the range from 0.02 to 0.03, the volumetric coefficient of moisture expansion could be interpreted to be between about 0.1 and 0.2. Using a result in this range for $\lambda_m$ in Equations (46) and (47) while assuming stiffness degradation primarily in the fiber/matrix interface, the predicted Von Mises strength for the matrix comes out to be nearly the same as that for the original dry coupons. Specifically, an expansion coefficient of 0.123 with a volumetric moisture concentration of 0.0256 from the experimental results gives the original dry Mises strength precisely.

Overall, between the observation of the matrix yield transition in the loading tests and the consideration of moisture-induced expansion, the predictions from the MCT-based model are more consistent with experimental observations when assuming that property degradation due to moisture absorption occurs primarily in the fiber/matrix interface, rather than throughout the bulk of the matrix. As this is the general consensus regarding the mechanism of moisture degradation, the application of MCT as a predictive tool in such contexts appears to be effective from the scope of the present work.

## 5. Conclusions

The experimental testing results revealed a degradation in the stiffness and strength of the carbon–glass hybrid fiber-reinforced composite coupons, mainly in the transverse direction. Only minimal degradation was seen in the mechanical performance in the longitudinal direction. These results are consistent with previous findings and suggest that the breakdown in mechanical properties due to moisture absorption in FRPs is primarily associated with the resin matrix. The properties measured in the dry coupons were generally consistent with the values predicted by models using published data for constituent materials. The constituent stresses at the loading of ultimate failure calculated using MCT

were also generally consistent with the published data. The modeling results were most consistent with experimental observations when guided by the established assumption that the degradation of stiffness and strength from moisture absorption is mainly localized at the fiber/matrix interface. A follow-up study to directly verify this assumption with visual scanning on coupons with identical layup would be of benefit.

The MCT approach was shown as a potentially effective tool for predicting performance and failure in structural analysis involving complex composites, and accounting for environmental effects accurately in a physically meaningful way. Micromechanics modeling and sensitivity analysis can be a powerful tool in predicting complex responses and verifying properties that might otherwise be difficult to obtain. Structural analysis tools with exact adjoint-based sensitivity capabilities are valuable assets in performing these analyses efficiently. These tools can be adopted by marine energy developers to help predict operation and maintenance costs for marine energy devices in their effort to reduce the levelized cost of energy for these devices.

**Author Contributions:** Data curation: B.G., J.N. and M.I.; Formal analysis: E.A.; Funding acquisition: B.G. and B.A.H.-S.; Investigation: J.N. and M.I.; Methodology: E.A.; Project administration: B.A.H.-S.; Resources: B.G.; Software: E.A.; Supervision: B.G. and B.A.H.-S.; Writing—original draft: E.A., B.G., J.N., M.I. and B.A.H.-S.; Writing—review and editing: E.A. and B.G. All authors have read and agreed to the published version of the manuscript.

**Funding:** This work was funded by the US Department of Energy, EERE Water Power Technologies Program. Sandia National Laboratories is a multi-mission laboratory managed and operated by National Technology and Engineering Solutions of Sandia, LLC., a wholly owned subsidiary of Honeywell International, Inc., for the U.S. Department of Energy's National Nuclear Security Administration under contract DE-NA0003525.

**Institutional Review Board Statement:** Not applicable.

**Informed Consent Statement:** Not applicable.

**Data Availability Statement:** All experimental data used in the analysis or having any impact on the findings, results and conclusions presented in this article are included herein. All findings from previous works having influence on the presented material are cited and listed in the References section.

**Conflicts of Interest:** The authors declare no conflict of interest in the production of the contained publication. The funders had no role in the design of the study; in the collection, analyses, or interpretation of data; in the writing of the manuscript; or in the decision to publish the results.

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
