# Peer review of "A Multicontinuum-Theory-Based Approach to the Analysis of Fiber-Reinforced Polymer Composites with Degraded Stiffness and Strength Properties Due to Moisture Absorption"

_jmse, doi:10.3390/jmse11020421_

Round 1
Reviewer 1 Report
The paper deals with the effect of marine water absorption on the stiffness and strength of composite laminates. The paper is heavily based on a theory developed by Garnich et al [34,435} and is referred as multicontinuum theory (MCT). The originality of the paper lies in extending the MCT formulation to include moisture absorption. This is an interesting contribution for the readers.
The paper is well written and has sound theoretical background.
However there are a few points that deserved to be improved. The first one is the fact that the choice of strength test. The authors must clearly justify the choice of a bearing test rather than standard tensile tests that are usually used in the literature. The bearing test is a complex non-linear problem that introduces a strong stress concentration in the laminate. From figure 5 it is clear that all specimens failed at the bolt hole. Of course there are several practical cases that justify this choice but the type of failure of the tests must also be included in the discussion.
Another point is the fact that a few test must be performed with dry and conditioned specimens to calibrate the model. From the results, it seems that the the stiffness degradation could successfully be calibrated in the numerical model.
However, the strength of a composite laminate is much more complex than the stiffness particularly if the laminate is not unidirectional. It is not clear in the text which criterion that was used to numerically evaluate the laminate strength.
It was also important to remark in the paper that a composite strength is highly dependent on the quality of the manufacturing. This implies that the model does require calibration tests for every different manufacturer and different manufacturing procedure and even stress concentration types.
This reviewer recommends the paper to be accepted provided that the authors properly address the points raised above.
Author Response
Thank you for your feedback. With regard to the first point, it's true the choice of strength test is significant and worth discussing. We have added some explanation at the end of section 2.2 to clarify this. In short, the bearing test configuration was chosen because it reflects the most common fabrication methods used in the marine industry. Such tests are apt to give results which directly correlate to application in practice.
Immediately following are some comments regarding the points on calibration/manfacturing considerations. If we take your meaning correctly, this idea of calibrating the MCT model was a large part of what we aimed to accomplish in this work. We see the experimental tests and modeling efforts as working together to understand effective properties and how they change with environment, which indeed can be a function of manufacturing/methods.
As a point of clarification regarding the strength of a laminate, in this work we did not employ a criterion such as Puck or LaRC as mentioned in the introduction, but with MCT analysis in which onset of failure is predicted when a constituent material reaches its ultimate limit. In this case the Mises stress of the carbon fiber and epoxy matrix are used as the driving criteria. There is also more discussion of the observed failure in the tested coupons at the beginning of Section 4.
Thanks again for your review.
Reviewer 2 Report
The manuscript is very well structured, the presentation of materials and methods is very clear, and the results are consistent with the measurements performed. There are however some minor details that should be corrected before the manuscript is ready for publication.
1.- p. 4 / line 160: equations (4) and (5) are cited, but actually equations (3) and (4) should be written instead.
2.- p. 6 / line 202: write in parentheses the temperature in degrees Celsius.
3.- p. 9 / line 296: give details about the AStrO software, whether it is a proprietary software or a publicly available code, where it can be obtained and what features it has.
However, there is a more fundamental question that appears at different times throughout the manuscript (see, for example, p. 18 / line 558-561) and in the conclusions. It is about the change in the behavior of the matrix (evident from the change in slope seen in the graphs shown in Figure 6) and, crucially, the mechanisms behind this change in properties. The arguments given to justify that it is the fiber-matrix interface that degrades (not the bulk polymeric matrix) are very weak, or at least need a more convincing explanation. No images of the microstructure are presented to support that the mechanisms suggested by the authors are correct. It seems that the entire justification relies on "This hypothesis has been indicated in previous studies as well." A microscopic study should be added or at least support in more detail the proposed interface degradation mechanisms.
Otherwise, I am of the opinion that the manuscript is adequate for the contents of the Journal and should be published after making these suggested minor improvements.
Author Response
Thank you for your thoughtful feedback. The minor line corrections have been made as suggested, with a reference and further explanation of the AStrO software. The package has been recently renamed ASenD3D, and the public repository is still under development, but it is in fact open source and accessible at the provided URL, https://github.com/MSDOToolz/ASenD3D.
Your point regarding the mechanisms of matrix degradation is well taken. We would love to produce microstructural scans or other imaging to verify directly that the breakdown is heavily localized in the fiber-matrix interface as further substantiation. Regrettably we do not possess any results of this kind for any of the samples tested herein, as such steps were not allocated for in the testing phase of this work. A short suggestion of this as a potential follow-up work was added.
However, for the present study the aim was not so much to substantiate or prove the microstructural mechanisms of degradation per se, but to validate MCT as an effective predictive tool when extended into this context of species absorption and property degradation. We are more or less taking it as a given that the degradation is interface-driven, as described and indicated in previous works, and showing that our predictions assuming that are most in line with what is observed.
Perhaps the discussion of this came across as circular reasoning initially, seeming to use the model to validate the mechanism and the mechanism to validate the model, when it is mainly just the latter we are driving at. We have re-worked some of the language around this, to hopefully clarify that context.
Thanks again for your review.